# Design of and Experiment on Reciprocating Inter-Row Weeding Machine for Strip-Seeded Rice

**Yu Wang** [1], **Xiaobo Xi** [1,*], **Meng Chen** [2], **Yangjie Shi** [1], **Yifu Zhang** [1], **Baofeng Zhang** [1], **Jiwei Qu** [1] **and Ruihong Zhang** [1]

1   School of Mechanical Engineering, Yangzhou University, Yangzhou 225000, China
2   Ningbo Fotile Kitchen Ware Company, Ningbo 315336, China
*   Correspondence: xbxi@yzu.edu.cn

**Abstract:** To solve the problems of high labor costs, a low weeding rate and a high seedling injury rate in the direct seeding of rice fields, this paper presents a reciprocating inter-row weeding machine for strip-seeded rice. The machine uses a combination of weeding wheels and weeding shovels to improve the efficiency of weeding between rice rows. Its reciprocating mechanism was designed and optimized. The simulation model of weeding teeth–paddy soil interaction was established in EDEM. The structural parameters of the weeding teeth were optimized, and the bending angle of the optimized weeding teeth was 55°. A prototype trial production and field tests were carried out. The results showed that the prototype's inter-row weeding rate was between 80.2% and 85.3% and the seedling injury rate was between 3.5% and 5.1% when the prototype's working speed was 1~3 km h$^{-1}$. The faster the speed of the prototype, the lower the inter-row weeding rate and the higher the seedling injury rate.

**Keywords:** agricultural machinery; paddy weeding machine; weeding shovel; reciprocating motion; EDEM

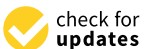



## 1. Introduction

China has a long history of the direct seeding of rice, which has been widely used by farmers because of its advantages in saving manpower and material resources [1]. Among direct seeding machines, rice strip-seeding machines are widely used for their high efficiency and convenient operation. However, a paddy field of strip-seeded rice is prone to problems, such as the high labor intensity of weeding and unsatisfactory weeding efficiency [2]. Weeds are a major constraint on rice yield and quality. Not only do they directly affect rice growth, but they are also a major host for rice pests and diseases [3–5]. At present, chemical weeding is the main method of weeding rice in China and is used in about 90% of the total planting area. Long-term, massive and high-frequency chemical weed control has led to increased weed resistance to herbicides, increased crop damage and increased environmental pollution [6–10]. As people become more health-conscious and environmentally conscious, their demand for pollution-free food is increasing. As a result, the use of chemical herbicides has been restricted, and more environmentally friendly and efficient physical weed control methods are being promoted.

Mechanical weeding is a kind of physical weeding in which mechanical parts interact with the soil and weeds to carry out the weeding process by such means as pulling out, pulling off or burying the weeds. In addition to destroying weeds, the mechanical weeding components can also turn the soil over, increasing the concentration of oxygen in the soil and promoting the absorption of nutrients by the crop [11]. Many small and medium-sized enterprises in China have begun to develop inter-row weeding machines, but the weeding efficiency of these machines needs to be further improved.

At present, many scholars and enterprises have developed a variety of mechanized weeding equipment for paddy fields. Guixiang Tao et al. [12] developed a weeding machine which drives a steel wire flexible shaft with a movable spring-tooth claw disk to

weed, which had a good weeding effect and a low seedling injury rate. Xu Ma et al. [13,14] developed the rake-tooth weeding wheel, which improved the weeding performance of the weeding wheel. Chongyou Wu et al. [15] developed the 2BYS-6 type paddy weeding machine, which could weed simultaneously between rows and plants in the field. Zhongzhe Li [16] developed the 3ZS-l paddy weeding machine, which was light, compact and easy to operate. The MSJ-4 rice field weeding machine from Japan could be flexibly steered at the ground to reduce seedling injury [17]. The three-wheeled riding paddy weeding machine from Seiyaku Co. used a feather-wheel structure as an inter-plant weeding component, which could reduce seedling injury significantly [18]. The Japanese Kubota's SJ-8N paddy weeding machine could simultaneously weed between rows and plants and had a long working width to improve weeding efficiency [19].

The above-mentioned research work mostly concerns weeding methods, the structural design of weeding components and the way the machine travels. The weeding effect of the machine can be further improved, however, and the problem of weeding omissions still not well solved yet. For this paper, a reciprocating inter-row paddy weeding machine for strip-seeded rice was designed, and the prototype was tested in the field. The reciprocating inter-row paddy weeding machine for strip-seeded rice can provide a new reference for the design of mechanized weeding equipment for paddy fields.

## 2. Materials and Methods

### 2.1. Machine Design

#### 2.1.1. Overall Structure

The overall structure of the reciprocating inter-row paddy weeding machine for strip-seeded rice is shown in Figure 1. The structure includes rake weeding wheels, weeding shovels, eccentric wheels, frames, springs, drive shafts and so on. The weeding work of the machine is mainly carried out by weeding shovels and weeding wheels. The power input shaft and transmission box are used to transmit the power from the tractor to the machine. The eccentric wheels are mounted on the drive shaft. Between the weeding wheels and the weeding shovels, and between the front and rear weeding shovels, springs are installed to facilitate the reciprocating motion of the weeding shovels. The weeding shovels, weeding wheels, springs and frame form a profiling mechanism that reduces the effect of ground heaving on the machine, which can maintain the stability of the machine when weeding. In addition, the machine is used in combination with a double axis rotary tillage fertilizing and land seeding compound operation machine [20]. The double axis rotary tillage fertilizing and land seeding compound operation machine would make two drains in the field while working. For this reason, there is a large gap between the weeding shovel in the middle and the weeding shovel on the sides.

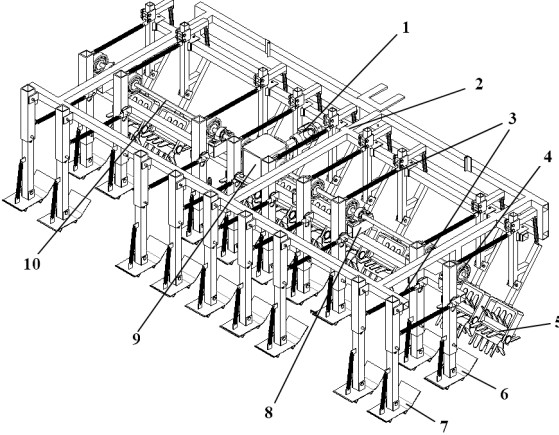

**Figure 1.** Overall structure of reciprocating inter-row paddy weeding machine for strip-seeded rice. 1. Power input shaft 2. Frame 3. Spring 4. Eccentric wheel 5. Rake weeding wheel 6. Front weeding shovel 7. Rear weeding shovel 8. Soleplate 9. Transmission box 10. Transmission shaft.

### 2.1.2. Working Principle

The reciprocating inter-row paddy weeding machine for strip-seeded rice improves the efficiency of weeding by combining passive weeding and active weeding. The rake weeding wheels are passively rotated due to soil action and rake down weeds along the way. The tractor transmits power to the transmission shaft through the gear structure in the transmission box, which make the transmission shaft rotate. The rotation of the drive shaft drives the eccentric wheels, which are mounted on the drive shaft. As a result, the front weeding shovels, which are immediately adjacent to the eccentric wheels, move reciprocally under the action of the eccentric wheels. Then, the front weeding shovels transmit their motion state to the rear weeding shovels through springs to make them reciprocate, too. The front and rear weeding shovels carry out their weeding operations by repeatedly crushing weeds and disturbing the soil. Technical details are listed in Table 1.

**Table 1.** Performance parameter table of reciprocating inter-row paddy weeding machine for strip-seeded rice.

| Items | Technical Parameter |
|---|---|
| Supporting power | 11.3–13.3 kW |
| Number of weeding rows | 9 |
| Working speed | 1.8–2.5 km·h$^{-1}$ |
| Working width | 3 m |
| Spacing of weeding components | 300/600 mm |
| Height of frame above ground | 580–620 mm |
| Overall weeding depth | 40–80 mm |
| Single row weeding width | 150 mm |

### 2.1.3. Weeding Wheel

As shown in Figure 2, a weeding wheel is mainly composed of a roller, a support bar, a connecting rod, a soleplate, a spring and a rake weeding wheel. The connecting rod, spring and strut form a profiling structure. When the machine travels across sunken ground, the springs extend, so that the weeding wheels will not immediately sink and cause congestion. When the machine travels over slabbed or bulging ground, the springs contract to allow the weeding wheels to break through the compacted or raised soil. The profiling structure allows the wheels to move forward without excessive undulation, stabilizing the weeding depth when the wheels are working.

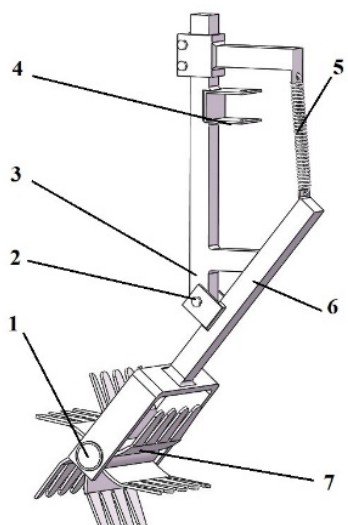

**Figure 2.** Structure diagram of weeding wheel. 1. Roller 2. Bolted connection point 3. Support bar 4. Soleplate 5. Spring 6. Connecting rod 7. Rake weeding wheel.

The rolling radius of the rake weeding wheel in an ideal state (no slippage) $r_y$ is determined by Equation (1).

$$r_y = a + c + l \tag{1}$$

where $a$ is the height of the weeding wheel roller measured to the bottom of connecting rod; $c$ is the distance from the bottom of connecting rod to the surface of the mud; $l$ is the depth to which the weeding wheel rake penetrates into the ground.

Figure 3 shows an analysis of the weeding wheel's movement under ideal conditions. To reduce structural wear and ensure the quality of the weeding, the rake wheel should be kept in a rolling state as much as possible when weeding in the field. However, due to changing soil conditions, the weeding wheel will inevitably slip [21] (horizontal travel speed is less than the linear speed of the weeding wheel rotation), and the degree of slip can be expressed as the slip rate $\delta$, as determined by Equations (2)–(5).

$$\delta = \frac{v_l - v}{v_l} \tag{2}$$

$$v_l = r_y \omega \tag{3}$$

$$v = r\omega \tag{4}$$

$$r = \frac{S}{2\pi} \tag{5}$$

where $v_l$ is the linear speed of the rake tines of the weeding wheel at point B under ideal conditions; $v$ is the actual speed of the weeding wheel (the translational speed of the weeding wheel center); $\omega$ is the angular speed of the weeding wheel; $r$ is the actual rolling radius of the weeding wheel; $S$ is the distance that the machine moves when the weeding wheel rotates for one revolution.

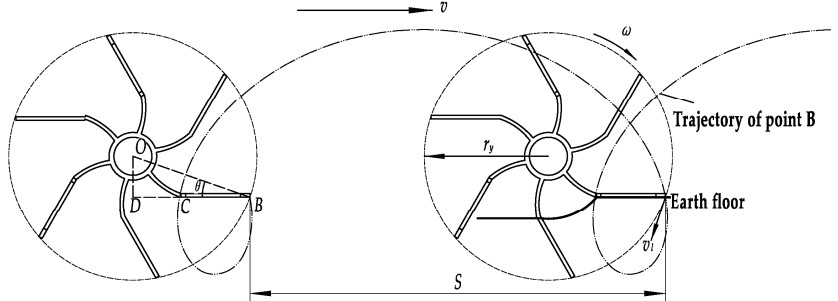

**Figure 3.** Motion analysis diagram of weeding wheel under ideal state.

From Formulas (2)–(5) we can obtain

$$S = 2\pi r_y (1 - \delta) \tag{6}$$

In order to reduce the leakage rate, the sum of the length of all of the rake tines should be greater than or equal to the travel distance of the machine when the weeding wheel rotates one turn (under ideal conditions) [22], which could be expressed as Equation (7).

$$bZ \geq 2\pi r_y (1 - \delta) \tag{7}$$

where $b$ is the length of straight section $BC$ of the rake tine and $Z$ is the number of rake tines. To ensure that the weeding wheel will not become clogged with mud while working, the number of rake tines should not be too great.

When the moving speed of the machine is 1~3 km h$^{-1}$, the rotation speed of the weeding wheel is 70~150 r min$^{-1}$. This can be used to find the distance $S$ and the length of the straight section $BC$. To improve the working efficiency of the rake tines, the straight section of the rake tines $BC$ should fit the mud surface when entering the soil, which means

that △ODB is a right-angle triangle. The length of OD is known to be the sum of *a* and *c*. Then, based on the above information, the degree of bending angle θ of the rake tines can be found.

### 2.1.4. Weeding Shovel

The front and rear weeding shovels of the reciprocating inter-row paddy weeding machine for strip-seeded rice are identical in structure. The weeding shovel is consisted of a weeding board, a connecting bar and a spring, as shown in Figure 4. The weeding shovel, connecting rod and spring formed a profiling structure. When the machine travels over slabbed, bulging or sunken ground, the spring can stabilize the weeding shovel by contracting or stretching to maintain a stable weeding depth. To reduce the resistance of the weeding shovel, the front part of the weeding shovel is designed with a slide structure. To improve the weeding efficiency, three rows of weeding teeth are set at the bottom of the weeding board, and the teeth are placed in staggered rows, as shown in Figure 5.

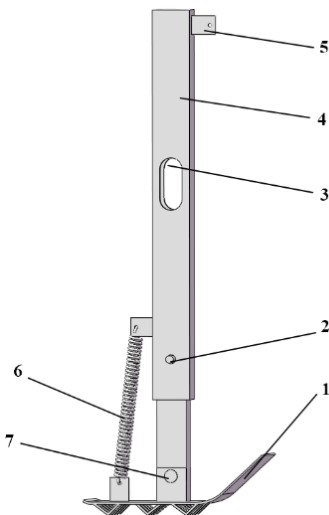

**Figure 4.** Structure drawing of weeding shovel. 1. Weeding board 2. Connection hole 3. Adjustment hole 4. Support bar 5. Ear plate 6. Spring 7. Mounting hole.

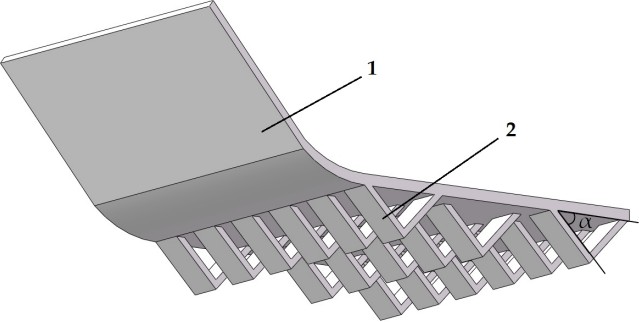

**Figure 5.** Bottom structure of weeding board. 1. Slide structure 2. Weeding teeth.

The eccentric wheel rotates, driving the weeding shovel immediately adjacent to the eccentric wheel in a reciprocating motion. Due to this reciprocating motion, the trace of the shovel will have a repetition area. The larger the repetition area, the better the weeding shovel's soil-disturbing ability and the stronger the weeding shovel's weeding ability. Usually, the lower the machine's moving speed, the greater the frequency of the reciprocating motion of the weeding shovel, and the greater the amplitude of the pendulum swing. The greater the amplitude of the pendulum swing, the larger the repetition area. According to the design manual [23], when the moving speed is 0.5~0.8 m s$^{-1}$, the swing

frequency of the pendulum is 210 min$^{-1}$, the swing amplitude is 140 mm and the overlap is 10 mm, the loosening and weeding ability of the reciprocating working parts is stronger. In order for the weeding shovel to weed evenly, the front and rear swing amplitude *GF* and *FJ* should be equal. This requires the weeding shovel to be vertical when *NM* is in the vertical state, and the values of d and $R_k$ are equal at this time. The eccentric wheel is mounted on the transmission shaft. Considering that the transmission box should be installed on the center of the frame and should not be too large, the values of $l_1$, $l_2$, $l_3$ and *d* can be designed accordingly. Since the contact surface of the eccentric wheel and the pendulum is tangential, *NV* and *MT* are perpendicular to *EG* and *HJ*, respectively. The value of *NV* can then be obtained through trigonometry, from which the eccentric distance *e* of the eccentric wheel can be calculated. Figure 6 showed the movement state diagram of the weeding shovel.

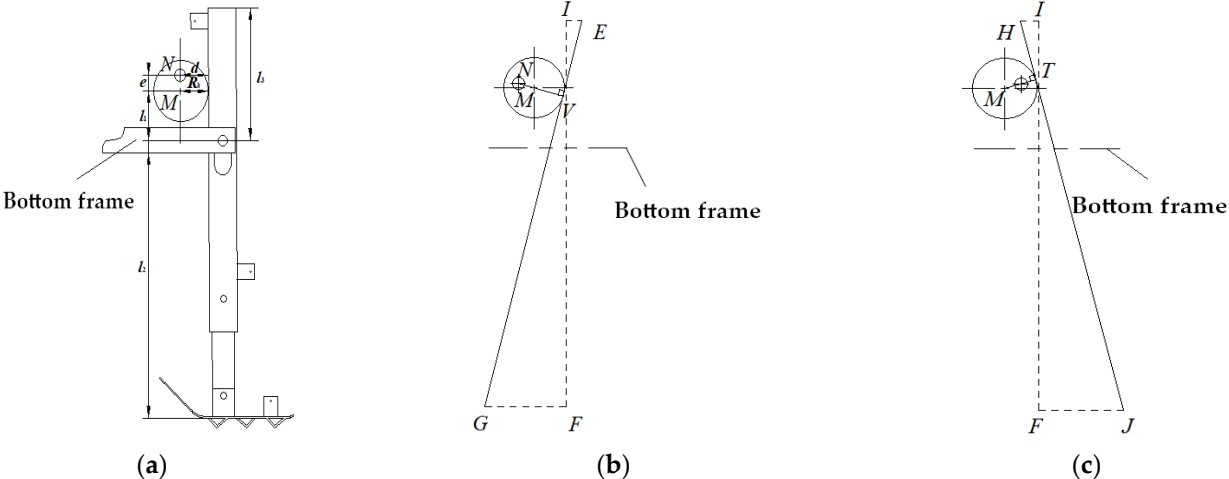

(**a**)　　　　　　　　　　　　　　　(**b**)　　　　　　　　　　　　　　　(**c**)

**Figure 6.** Movement state diagram of the weeding shovel: (**a**) vertical state, (**b**) forward swing state, (**c**) backward swing state. *N* is the center of eccentric wheel hub; *M* is the center of the eccentric contour; *I* and *F* are the two vertex positions of the swing rod in the vertical state; *E* and *G* are the two vertex positions of the swing rod when the swing rod swings forward to the farthest distance; *H* and *J* are the positions of the two vertices of the swing rod when the swing rod swings back to the farthest distance; *V* and *T* are the contact points between the eccentric wheel and the support bar when the swing rod swings forward and backward; *EG* and *HJ* are the straight lines in which the swing rod swings forward and backward; *d* is the distance from the center of the eccentric wheel hub to the swing rod when the weeding shovel is vertical; $R_k$ is the contour radius of the eccentric wheel; *e* is the eccentricity of eccentric wheel; $l_1$ is the distance from the center of the eccentric wheel hub to the bottom frame; $l_2$ is the distance from the bottom frame to the ground; $l_3$ is the distance between the bottom frame and the top frame.

## 2.2. EDEM-Based Optimization of the Bending Angle of the Weeding Teeth

The $\alpha$ in Figure 6 is the bending angle of the weeding teeth, the size of which will affect the soil entry resistance and the soil disturbance level of the weeding shovel. To optimize the weeding shovel, weeding teeth with different $\alpha$ angles were designed for simulation. To facilitate the production of weeding shovels, the angle of the weeding teeth was designed in 5° increments, and weeding teeth with $\alpha$ of 35°, 40°, 45°, 50°, 55°, 60°, 65°, 70° and 75° were designed for simulation experiments. The paper uses EDEM for discrete element simulation to analyze the effect of $\alpha$ on both the soil entry resistance of the weeding shovel and the level of soil disturbance by the weeding shovel when $\alpha$ is at different angles. Soil particles are disturbed by the weeding shovel and change their state of motion, and the level of soil disturbance can be identified by the number of moving particles and their moving speed [24].

The particle model uses Hertz–Mindlin with JKR Cohesion, which can simulate the physical properties of significant adhesion and agglomeration between soil particles in paddy fields due to electrostatic forces and moisture, among other things [25]. The paddy soil model can be divided into three layers: the upper layer is a layer of water particles, the middle layer is a layer of saturated mud where water particles are mixed with soil particles, and the lower layer is a layer of mud with a small amount of water particles. The working speed of the weeding machine in the paddy field is generally between 0.3 and 0.8 m s$^{-1}$. Therefore, in the simulation software, the swing angular speed of the weeding shovel was set to 90 r min$^{-1}$, and the moving speed of the weeding machine was set to 0.28, 0.56 and 0.83 m s$^{-1}$ for three sets of discrete element simulations. Loam was used as the research object in this study. A parametric model of the paddy soil was established in the EDEM, which is discrete element simulation software, according to the relevant references [26] and the data from the software's own soil model database. The paddy soil model parameters were determined as shown in Tables 2 and 3.

**Table 2.** Material parameter table.

| Parameters | Values |
|---|---|
| Density of soil | 1830 kg·m$^{-3}$ |
| Poisson's ratio of soil | 0.5 |
| Shear modulus of soil | 11.5 MPa |
| Radius of soil particle | 1 mm |
| Density of water | 1000 kg·m$^{-3}$ |
| Poisson's ratio of water | 0.5 |
| Shear modulus of water | 100 MPa |
| Radius of water particle | 0.5 mm |
| Density of steel | 7860 kg·m$^{-3}$ |
| Poisson's ratio of steel | 0.288 |
| Shear modulus of steel | 28,000 MPa |

**Table 3.** Contact parameters between materials.

| Materials | Recovery Coefficient | Static Friction Coefficient | Rolling Friction Coefficient |
|---|---|---|---|
| Water–Water | 0.05 | 0.05 | 0.01 |
| Soil–Soil | 0.05 | 0.90 | 0.01 |
| Soil–Water | 0.05 | 0.05 | 0.01 |
| Soil–Steel | 0.10 | 0.20 | 0.20 |
| Water–Steel | 0.10 | 0.20 | 0.20 |

*2.3. Field Experiment*

2.3.1. Working Environment and Agronomic Requirements

A diagram of the water field environment is shown in Figure 7. The soil of a paddy field could be divided into a mud layer and a clay layer. The depth of the mud layer is 30–50 mm, and the depth of the soil layer is 160–180 mm. Three weeks after rice seeds have been sown, the root distribution depths of the rice seedlings and weeds are 80–100 mm and 30–50 mm, respectively. The distance between rice rows is 250–300 mm. According to the Technical Specification for the Evaluation of the Quality of Seedling Weeding Machine (DB23/T930-2005), the weeding rate between rows should be greater than 80%, and the seedling injury rate should be less than 5% [27].

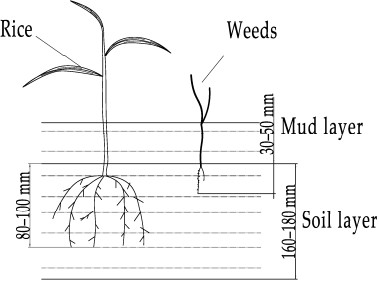

**Figure 7.** Diagram of water field environment.

2.3.2. Experimental Trial of Weeding Performance

The experiment was carried out on 29 June 2021 in the experimental field of Jin Yun Agricultural Technology Development Co Ltd., Jiangdu District, Yangzhou City, Jiangsu Province (119.512° E, 32.562° N). The rice in the experimental area was strip seeded by a rice direct seeding machine. The rice seedlings' height was 200–300 mm, the row spacing was 250 mm, and the weeds' height was 100–200 mm. The speed of the weeding machine was an important factor affecting the weeding effect, so a single-factor test was conducted, with the speed of the machine as the test factor and the weeding rate and seedling injury rate as the evaluation indices. The test plot was divided into six zones, each 2.5 m × 100 m in size. Five areas within each of the six zones were randomly selected as the test areas, and each of them was 2.5 m × 1 m.

The indicators set for the experiment were inter-row weeding rate $\lambda$ and seedling injury rate $\eta$, which can be expressed as Equations (7) and (8):

$$\lambda = \frac{N_b - N_a}{N_b} \times 100\% \tag{8}$$

$$\eta = \frac{D_b}{D_a} \times 100\% \tag{9}$$

where $N_b$ is the number of weeds in the test area before weeding. $N_a$ is the number of weeds in the test area after weeding. $D_b$ is the number of damaged rice seedlings in the test area (broken stalks, bent stalks and damaged epidermis). $D_a$ is the number of rice seedlings in the test area.

In the experiment, the prototype was operated at 1 km h$^{-1}$, 2 km h$^{-1}$ and 3 km h$^{-1}$ in six zones, and the experiment was repeated twice at each speed. The weed condition and rice condition in the test areas were recorded, the inter-row weeding rate and seedling injury rate were calculated for statistical purposes and the average value was calculated. The weeding rate and seedling injury rate were statistically analyzed by ANOVA with the significance level $\alpha$ = 0.05.

## 3. Results and Discussion

### 3.1. Results of Simulation

With EDEM post-processing, the total number of soil particles with different moving velocities when the weeding shovel moved to the lowest position could be counted. At the three moving speeds of the weeding shovel, the moving speeds of soil particles ranged from 0 to 3.5 m s$^{-1}$, and the moving speeds of soil particles that were disturbed due to contact with the weeding shovel ranged from 0.12 to 3.2 m s$^{-1}$. Therefore, the particles with a speed greater than 0.12 m s$^{-1}$ were specified as disturbed soil particles, and the statistical results of disturbed soil particles are shown in Table 4. According to the table, the degree of soil disturbance by the weeding shovel was greater when the $\alpha$ angle was 45°, 50° and 55°, with the best soil disturbance occurring when $\alpha$ was 45°, followed by 50° and 55°.

**Table 4.** Table of statistical results of disturbed particles.

| α Angle | Count of Disturbed Particles | | |
|---|---|---|---|
| | *v* = 0.28 | *v* = 0.56 | *v* = 0.83 |
| 30° | 21,989 | 22,920 | 15,947 |
| 35° | 26,161 | 23,814 | 17,667 |
| 40° | 26,989 | 24,209 | 20,425 |
| 45° | 32,617 | 27,573 | 26,289 |
| 50° | 28,832 | 26,229 | 22,701 |
| 55° | 27,174 | 25,133 | 21,876 |
| 60° | 23,688 | 22,129 | 19,884 |
| 65° | 22,545 | 21,043 | 18,257 |
| 70° | 21,087 | 20,881 | 18,171 |
| 75° | 20,190 | 19,577 | 17,334 |

With EDEM post-processing, the resistance of the weeding shovel in disturbing the soil at three forward speeds can be obtained. Figure 8 shows the weeding shovel resistance corresponding to different α angles, and Table 5 shows the maximum and average values of resistance corresponding to the different α angles of the weeding teeth. As can be seen from the table and figure, the resistance of the weeding shovel when entering the soil is low when α is 55°, 60° and 65°, and the resistance is the least when the α is 55°, followed by 60° and 65°.

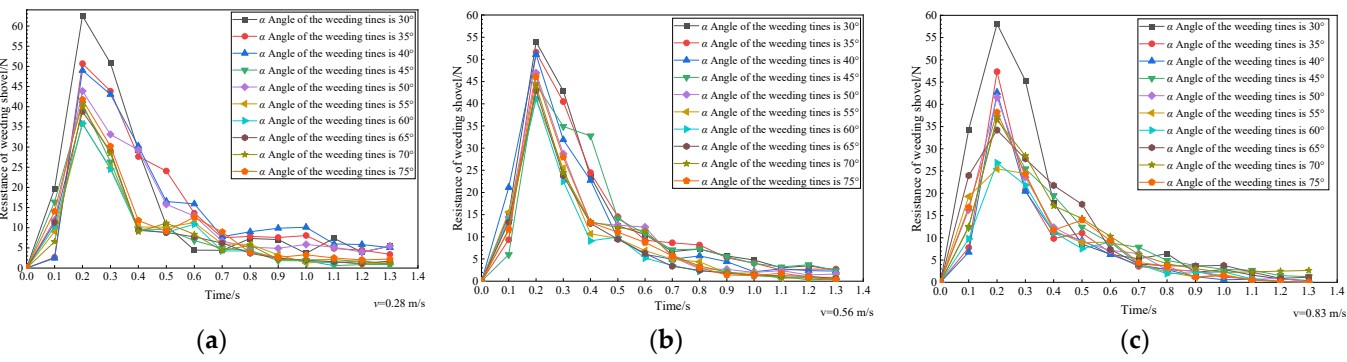

**Figure 8.** Simulation results of resistance of weeding shovel. (**a**) When the moving speed of weeding shovel is 0.28, the resistance of weeding shovel with different α angle. (**b**) When the moving speed of weeding shovel is 0.28, the resistance of weeding shovel with different α angle. (**c**) When the moving speed of weeding shovel is 0.28, the resistance of weeding shovel with different α angle.

**Table 5.** Statistical table of resistance of weeding shovel.

| α Angle | Maximum Resistance | | | Average Resistance | | |
|---|---|---|---|---|---|---|
| | v = 0.28 | v = 0.56 | v = 0.83 | v = 0.28 | v = 0.56 | v = 0.83 |
| 30° | 62.469 N | 53.931 N | 58.043 N | 15.451 N | 13.707 N | 13.567 N |
| 35° | 50.653 N | 51.607 N | 47.342 N | 14.715 N | 12.131 N | 10.038 N |
| 40° | 48.987 N | 51.109 N | 42.674 N | 13.649 N | 11.941 N | 8.179 N |
| 45° | 40.740 N | 44.401 N | 37.026 N | 13.133 N | 11.569 N | 8.918 N |
| 50° | 43.866 N | 46.940 N | 41.573 N | 9.896 N | 10.527 N | 7.589 N |
| 55° | 35.722 N | 41.075 N | 25.489 N | 8.332 N | 8.138 N | 6.985 N |
| 60° | 35.877 N | 42.155 N | 26.840 N | 8.714 N | 8.695 N | 7.773 N |
| 65° | 38.800 N | 43.977 N | 34.216 N | 8.813 N | 9.039 N | 8.993 N |
| 70° | 39.984 N | 44.055 N | 36.476 N | 9.729 N | 9.379 N | 10.874 N |
| 75° | 41.740 N | 46.082 N | 38.197 N | 10.349 N | 9.523 N | 10.183 N |

According to the above analysis of the disturbed soil particles and the weeding shovel's working resistance, when the α angle of weeding teeth is 55°, the working resistance of

the weeding shovel is small and its soil-disturbing level is great, making this the preferred design angle.

### 3.2. Results of Field Experiment

Table 6 shows the statistical data of the inter-row weeding rate and seedling injury rate of the machine at different speeds. The inter-row weeding rate of the machine ranged from 80.2 to 85.3% at speeds of 1 to 3 km h$^{-1}$, meaning it had a higher weeding rate than traditional wheeled and hand-held weeding machines [28–30]. At the same time, the increase in the weeding rate of the machine necessitates a decrease in the weeding omission rate, indicating that the machine can lessen the problem of weeding omission.

**Table 6.** Inter-row weeding rate and seedling injury rate of machine at different speeds.

| Speed | 1 km·h$^{-1}$ | 2 km·h$^{-1}$ | 3 km·h$^{-1}$ |
|---|---|---|---|
| Inter-row weeding rate | 85.3 ± 2.2% | 82.7 ± 1.2% | 80.2 ± 1.5% |
| Seedling injury rate | 3.5 ± 0.7% | 4.2 ± 0.5% | 5.1 ± 0.8% |

Table 7 shows the ANOVA of the influence of weeding speed on the inter-row weeding rate. The *p*-value was smaller than 0.05, indicating that the machine's moving speed has a significant effect on the weeding rate between rows. When the moving speed of the machine increases, the inter-row weeding rate of the machine decreases. The reason for this is that, when the speed of the machine increases, the slip rate of the weeding assembly increases during weeding, which prevents the weeding assembly from adequately mowing or pulling weeds, resulting in a decrease in the inter-row weeding rate [31,32].

**Table 7.** Variance analysis of influence of weeding operation speed on inter-row weeding rate.

| Source | SS | DF | MS | F | *p*-Value | F-Crit |
|---|---|---|---|---|---|---|
| Between Groups | 39.02 | 2 | 19.51 | 6.68 | 0.028 | 5.143 |
| Within Groups | 17.06 | 6 | 2.84 | | | |
| Total | 56.08 | 8 | | | | |

The injury rate of the machine was between 3.5 and 5.1% when the machine was in the speed range of 1 to 3 km h$^{-1}$, which met the inter-row weeding rate requirement of rice weeding. Table 8 shows the ANOVA of the influence of weeding speed on seedling injury rate. The *p*-value was less than 0.05, indicating that speed has a significant effect on the seedling injury rate. When the machine's speed increases, the injury rate increases. The reason for this is that, when the speed of the machine increases, the tractor operators become less precise in their operations, which leaves the machine prone to drifting in the field [33,34]. Therefore, in order to meet the agronomic requirements of inter-row weeding, the speed of the machine should not be too high.

**Table 8.** Variance analysis of influence of weeding operation speed on seedling injury rate.

| Source | SS | DF | MS | F | *p*-Value | F-Crit |
|---|---|---|---|---|---|---|
| Between Groups | 3.86 | 2 | 1.93 | 5.26 | 0.047 | 5.143 |
| Within Groups | 2.2 | 6 | 0.37 | | | |
| Total | 7.28 | 8 | | | | |

Considering that the travel speed of 1 km h$^{-1}$ was too slow, the minimum working speed of the machine was set at 1.8 km h$^{-1}$. When the machine was working at 3 km h$^{-1}$, the injury rate of the machine reached 5.1%, which exceeded the agronomic requirement of 5%, so the maximum moving speed of the machine was set at 2.5 km h$^{-1}$ to keep the injury rate of the machine within the specified range.

Figure 9 shows the weeding effect of the reciprocating inter-row paddy weeding machine for strip-seeded rice. There were no obvious weed residues in the worked area, and the damage to the rice seedlings was light, which meets the agronomic requirements of inter-row rice weeding and also shows that the machine performed well.

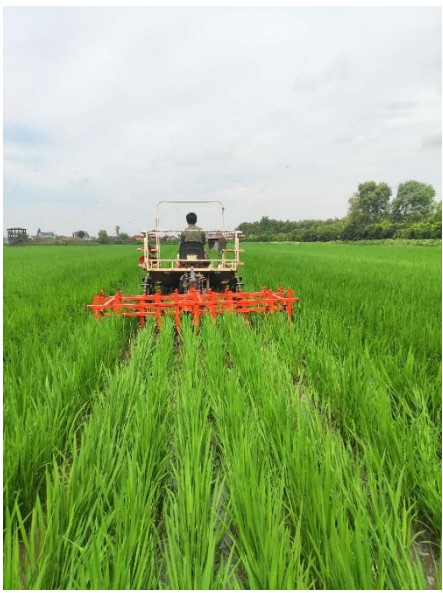

**Figure 9.** Weeding effect of reciprocating inter-row paddy weeding machine for strip-seeded rice.

## 4. Conclusions

To solve the problems of high labor costs, a low weeding rate and a high seedling injury rate in the direct seeding of rice fields, an inter-row paddy weeding machine for strip-seeded rice was designed. The machine uses a combination of weeding wheels and weeding shovels to weed between the rows. The structure of the weeding wheel was designed based on its working principle and the weeding requirements in the field. Parameters such as the width of the weeding wheel and the number of rake teeth have been determined according to the agronomic requirements for weeding. Parameters such as the length of the rake teeth of the weeding wheel and its bending angle have been calculated based on the analysis of the movement of the weeding wheel. The reciprocating mechanism was designed and optimize based on the working principle of the weeding spade and the weeding requirements of the field. The width of the weeding shovel and the arrangement of the weeding tines have been determined according to the agronomic requirements for weeding. The frequency and amplitude of oscillation of the weeding shovel can be determined according to the agronomic design handbook in order to calculate parameters such as the installation position of the eccentric wheel and its eccentric distance. This paper uses EDEM to develop a simulation model of weed control teeth–paddy soil interaction. Discrete element experiments were conducted on different weeding tine bending angles to find the optimal bending angle, thus improving the efficiency of the weeding shovel. When the $\alpha$ angle was 55°, the weeding shovels worked with less resistance and better soil disturbance. Field experiments were carried out with the prototype machine. The results show that, when the machine is operated at a speed of 1 to 3 km h$^{-1}$, the weeding rate between rows ranges from 80.2% to 85.3%, and the seedling injury rate ranges from 3.5% to 5.1%. The results of the experiment meet the agronomic requirements for rice weeding in the field. The experiments have also shown that the traveling speed of the machine has a significant effect on the weeding rate and seedling injury rate: the faster the traveling speed, the lower the weeding rate and the higher the seedling injury rate.

Mechanical weeding can avoid the use of herbicides and can greatly reduce the ecological impact of rice cultivation on the farmland environment, which is of great significance

to agricultural production. The reciprocating inter-row paddy weeding machine for strip-seeded rice designed in this paper can ensure the desired weeding effect while maintaining a low seedling injury rate, and its design can serve as a reference for the development of paddy weeding machines.

**Author Contributions:** Conceptualization, Y.W. and X.X.; methodology, Y.W., X.X., Y.S. and Y.Z.; cc, Y.W., X.X., Y.S. and Y.Z.; formal analysis, Y.W., Y.S., B.Z. and J.Q.; investigation, Y.W., Y.S., B.Z. and J.Q.; data curation, Y.W. and M.C.; writing—original draft preparation, Y.W.; writing—review and editing, Y.W. and X.X.; visualization, X.X. and R.Z.; supervision, X.X. and R.Z.; project administration, B.Z. and R.Z.; funding acquisition, X.X. and R.Z. All authors have read and agreed to the published version of the manuscript.

**Funding:** This research was funded by Yangzhou University Interdisciplinary Research Foundation for Crop Science Discipline of Targeted Support (yzuxk202007), Jiangsu Agricultural Science and Technology Innovation Fund (CX(22)3098), Jiangsu Modern Agricultural Machinery Equipment and Technology Demonstration and Promotion Project (NJ2021-65) and High-end Talent Support Program of Yangzhou University.

**Institutional Review Board Statement:** Not applicable.

**Informed Consent Statement:** Not applicable.

**Data Availability Statement:** Not applicable.

**Acknowledgments:** The authors would like to thank the technical support of their teacher and supervisor. We also appreciate the assistance provided by team members during the experiments. Moreover, we would like to thank Yangzhou Huilong Machinery Manufacturing Co., Ltd. for manufacturing the device. Additionally, we sincerely appreciate the work of the editor and the reviewers of the present paper.

**Conflicts of Interest:** The authors declare no conflict of interest.

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
