# Peer review of "Design of and Experiment on Reciprocating Inter-Row Weeding Machine for Strip-Seeded Rice"

_agriculture, doi:10.3390/agriculture12111956_

Round 1

Reviewer 1 Report (Previous Reviewer 1)

This manuscript has been significantly improved and now warrants publication in agriculture.

Author Response

Thank you very much for your recognition. In response to your suggestions, this paper has been scientifically simplified in language and some spelling errors have also been corrected. Thank you again for your instructive comments.

Reviewer 2 Report (Previous Reviewer 2)

The quality of the revised paper has been improved. The structure of the article is clear, the data is detailed, the logic is more rigorous, and the quality of English has also been improved. However, there are still some problems in the article, which can be further improved.

1.The format of references is not standard, please revise the manuscript as required.

2.In order to complete the structure of the article, it is suggested to change the title of Chapter II to Materials and Methods.

3.Discussion is an important stage of paper writing, which can stimulate us to think about deeper reasons. It is suggested to discuss the results of simulation tests after Section 3.1.

4.In the discussion part of the test results, you can guess, but you also need theoretical support. It is suggested that the author add references to the discussion part after the experiment.

5.In the conclusion, the authors were asked to elaborate on how the reciprocating mechanism was designed and optimized.

6.The manuscript uploaded by the author has comments, so the author's format cannot be seen clearly. Please proofread by yourself.

Author Response

Reviewer 3 Report (Previous Reviewer 3)

Weeds in paddy fields are one of the main reasons for the decline of rice yield and quality. Effective weeding can provide good growth conditions for rice. In this study, a reciprocating inter-row paddy weeding machine for strip-seeded rice is designed. The reciprocating interrow paddy weeding machine for strip-seeded rice can provide a new reference for the design of mechanized weeding equipment for paddy fields.

Some reviewing comments about the content, structure and writing of this manuscript are listed below.

1. The author uses 1~3 km·h-1 as the working speed in the prototype test in the manuscript, but the working speed in Table 1 of the manuscript is 0.5-0.7 m·s-1, and this two parameters are different. The author should explain this.

2. In LINE 176, But the weeding effect of the machine can be further improved and the problem of weeding omissions still not well solved yet.According to the test results in this paper, weeding omission is still not well solved. Therefore, the author needs to refine the problems solved in this manuscript

3. In LINE 412 “horizontal travel speed was less than the linear speed of the weeding wheel rotation”, The reviewer wondered why the linear speed of the passive rotating rake weeding wheels was higher than the horizontal travel speed?

4. The reviewer's biggest doubt: in section 2.2, the author used EDEM software to simulate the test of weeding tee, but how to model water in EDEM? How are the physical parameters of water in the manuscript (as shown Shear modulus of water and Radius of water particle in Table 2) obtained? Water is not granular, can we use edem to simulate it? How to verify the simulation results?

5. The simulation contents and results in Section 3.1 are inconsistent with the field test contents and results in Section 3.2, so how to verify the simulation in Section 3.1? Verification of simulation results in Section 3.1 should be added to Section 3.2.

6. Other problems: the units of the same variable in the paper should be consistent.

Round 2

Reviewer 3 Report (Previous Reviewer 3)

The author has revised the paper, and the reviewers have no problems

This manuscript is a resubmission of an earlier submission. The following is a list of the peer review reports and author responses from that submission.

Round 1

Reviewer 1 Report

(1) The structure name that appears in "2.1.1. Overall structure" needs to be reflected in Figure 1, and the name needs to be unified, such as Line 76 “transmission box” is not reflected in Figure 1.

(2) The font format and size of all graphics in this paper need to be unified.

(3) English grammar and expression are not professional enough, please revise and polish.

(4) After2.1.3. Design of weeding wheels and 2.1.4. Design of weeding shovel, there is no design result parameter, please add and explan.

(5)What is the selection basis of “line176-177 ‘α of 45°, 50°, 55°, 60°, 176 65°, 70° and 75°’ ”?

(6)Line 188 “Loamy soils” is not accurate.

(7)line 207setup is not accurate。

(8)There's a speed difference between “line 187 ‘0.5 m/s’” and “line 212 is ‘1 km/h, 2 km/h and 3 km/h’ ”, so how do you make sure your simulation tests make sense?

Reviewer 2 Report

To solve the problems of high labor costs, low weeding rate and high seedling injury rate in direct seeding rice fields, an inter-row paddy weeding machine for strip seeding rice was designed. The machine uses a combination of weeding wheels and weeding shovels to improve the efficiency of weeding between rice rows. The reciprocating mechanism was designed and optimized. The simulation model of weeding shovel-paddy soil interaction was established in EDEM, and the structural parameters of weeding tines were optimized. Trial production and field experiment of the prototype were completed.

However, there are many mistakes in the manuscript and experimental results do not show good. The language of this manuscript needs some intensive improvement so that it can be more readable. The authors should be very careful about the grammar issues when writing. 

Specific comments:

1.Writing ideas are not clear, unable to highlight the focus of research.

2.Structural design is too simple, incomplete description.

3.Missing experimental data and inappropriate data processing.

4.Manuscripts are too vague and lacks theoretical depth.

Reviewer 3 Report

This study tries to uses a combination of weeding wheels and weeding shovels to improve the efficiency of weeding between rice rows. Although the topic is very interesting, it suffers from major limitations. There are some issues that should be addressed.

1. The manuscript lacks a condensation of innovation.

2. In LINE 89,“This machine improved the efficiency of weeding by combining passive weeding and active weeding.No evidence or results of improved the efficiency of weeding is provided in the manuscript.

3. Figure 1 is unclear and much of the information is not clearly represented, for example, the unclear expression of the structure of the eccentric wheel, and how the front weeding shovels reciprocate under the action of the eccentric wheel?

4. The parameter of the items “Overall weeding depth” in Table 1 should be a range.

5. In LINE 111 “The rolling radius of a rake weeding wheel in an ideal state (no slippage, no slippage)”??. In LINE 114 “a was determined to be 50 mm”, the author should explain the choice of parameters for a.

6. In LINE202” According to the agronomic requirements [27,28], the weeding rate between rows should be higher than 75% and the injury rate of seedlings should be less than 5%.” To the best of the reviewer's knowledge, these two references do not mention the agronomic requirements stated by the authors

7. I think it is necessary to compare the performance of the algorithm proposed in this manuscript with other existing algorithms proposed in many papers were published.

8. I did not find Discussion in this manuscript. Many papers were published in the field of paddy weeding machine. The author only briefly mentions the tests on the prototype and the results obtained, but there is no discussion of the results and no analysis of the reasons for the experimental results.